# Interprofessional and Intraprofessional Communication about Older People’s Medications across Transitions of Care

**DOI:** 10.3390/ijerph18083925

**Published:** 2021-04-08

**Authors:** Elizabeth Manias, Tracey Bucknall, Robyn Woodward-Kron, Carmel Hughes, Christine Jorm, Guncag Ozavci, Kathryn Joseph

**Affiliations:** 1School of Nursing and Midwifery, Centre for Quality and Patient Safety Research, Institute for Health Transformation, Faculty of Health, Deakin University, Burwood 3125, Australia; tracey.bucknall@deakin.edu.au (T.B.); gozavci@deakin.edu.au (G.O.); 2Deakin-Alfred Health Nursing Research Centre, Alfred Health, Melbourne 3004, Australia; 3Department of Medical Education, Melbourne Medical School, The University of Melbourne, Parkville 3010, Australia; robynwk@unimelb.edu.au; 4School of Pharmacy, Queen’s University Belfast, Belfast BT7 1NN, UK; c.hughes@qub.ac.uk; 5NSW Regional Health Partners, Wisteria House, James Fletcher Hospital, Newcastle 2300, Australia; Christine.Jorm@health.nsw.gov.au; 6School of Nursing and Midwifery, Faculty of Health, Deakin University, Burwood 3125, Australia; k.joseph@deakin.edu.au

**Keywords:** interprofessional relations, interprofessional communication, intraprofessional communication, medication communication, older people, transitions of care, health communication, medication therapy management, medication safety

## Abstract

Communication breakdowns contribute to medication incidents involving older people across transitions of care. The purpose of this paper is to examine how interprofessional and intraprofessional communication occurs in managing older patients’ medications across transitions of care in acute and geriatric rehabilitation settings. An ethnographic design was used with semi-structured interviews, observations and focus groups undertaken in an acute tertiary referral hospital and a geriatric rehabilitation facility. Communication to manage medications was influenced by the clinical context comprising the transferring setting (preparing for transfer), receiving setting (setting after transfer) and ‘real-time’ (simultaneous communication). Three themes reflected these clinical contexts: dissemination of medication information, safe continuation of medications and barriers to collaborative communication. In transferring settings, nurses and pharmacists anticipated communication breakdowns and initiated additional communication activities to ensure safe information transfer. In receiving settings, all health professionals contributed to facilitating safe continuation of medications. Although health professionals of different disciplines sometimes communicated with each other, communication mostly occurred between health professionals of the same discipline. Lack of communication with pharmacists occurred despite all health professionals acknowledging their important role. Greater levels of proactive preparation by health professionals prior to transfers would reduce opportunities for errors relating to continuation of medications.

## 1. Introduction

Medication incidents are common at transitions of care when patients move from one environment to another [1,2,3,4]. Medication incidents, which refer to errors or discrepancies in prescribing, preparing, administering or monitoring medications, are common in older people as they often have complex medication regimens. Older people regularly experience care transitions between the community, hospitals, rehabilitation facilities and residential aged care settings due to their changing health care needs.

There is a ‘usual practice’ for communicating across different transitions of care during the journey between home or aged care facilities, admission to hospital, transfers within hospital, discharge home or to aged care facilities and movements within community care facilities [2]. In the emergency department, prior to admission to hospital, a medication history is taken, either by a pharmacist or by an emergency doctor. The admitting doctor examines the medication history and medical record of the individual, including checking any previous admission to the hospital. Nurses complete a verbal handover between the emergency department and admitting ward. Discussions may also occur between the admitting doctor and senior consultant of the admitting ward or more junior medical staff may be involved. Pharmacists in the admitting ward are notified about a patient’s medication requirements by the admitting doctor’s written notes in the medical record and their notations in the medication chart. However, pharmacists tend to be assigned to medical specialties rather than specific wards. In communicating about a patient’s movements within a hospital, verbal handovers are usually conducted individually with single discipline groups comprising nurses, doctors or pharmacists. Following discharge from hospital, a discharge summary is written with information about the medications to be prescribed. Hospital doctors relay this discharge summary to the general practitioner or primary care doctor using secure electronic messaging systems or written letters. Facsimile use to convey information at the time of hospital discharge remains common in countries such as the United States, Australia and the United Kingdom [5,6].

The World Health Organization (WHO) developed the International High 5s Project to address issues involving patient safety. In regards to medication accuracy at transitions of care, the WHO also created a Standard Operating Protocol to provide health professionals globally with strategies to enhance communication about their medication activities [7]. The goal of this initiative was to reduce undocumented medication discrepancies and prevent medication-related harm as patients moved between settings.

Recent research examining medication management across transitions of care has focused on medication incidents [3,8], pharmacist-led interventions [9,10,11,12] and patient and family perspectives [13,14]. Research also identifies communication breakdowns contributing to medication incidents across transitions [15,16]; yet, little is known about what influences how health professionals communicate with each other to manage medications and avoid potential breakdowns in communication across transitions. Interview and focus group studies have investigated the perspectives of nurses, doctors and pharmacists [17] and communication between health professional disciplines [12,16,18]. Previous studies using observations that considered medication communication between health professionals were conducted within a single setting and examined general populations rather than focusing on older people [19,20,21].

Doctors, nurses and pharmacists comprise the three major health professional disciplines involved in medication management. Examining how communication occurs within and across these discipline groups to manage medications across transitions of care, can assist with understanding how communication contributes to medication incidents and to medication safety at transitions. The aim of this study was to examine how interprofessional and intraprofessional communication to manage older people’s medications across transitions of care occurred in acute and geriatric rehabilitation settings. Older people referred to individuals aged 65 years and over, and transitions of care included inter-ward, inter-facility and movements to and from community settings.

## 2. Materials and Methods

An ethnographic study was undertaken comprising observations, semi-structured interviews and focus groups.

### 2.1. Setting and Sample

This study was conducted across two sites in Melbourne, Australia: an acute tertiary referral hospital and a geriatric rehabilitation facility. Data were collected from two medical specialty wards, one general medical ward, two rehabilitation wards and three aged care wards.

Nurses, pharmacists and doctors employed on the selected study wards at least one day per week and who provided care to older patients were eligible for inclusion. Purposive sampling according to health discipline was used to determine possible participants. Participants were informed about the study through informal group or individual discussions in the clinical setting. Nurse managers assisted with identifying potential participants. Informed written consent was obtained from participants in interviews, observations and focus groups. Verbal consent was obtained from other health professionals who interacted with the formally consented participant during observations. With regards to other health professionals interacting with the formally consented participant, verbal consent was sought during information sessions before the conduct of observations. Alternatively, verbal consent was sought during the time of the observations, if those individuals were not available to attend information sessions. If a specific health professional chose not to have their conversations noted during observations, any recording of information involving that person immediately ceased. Observations were scheduled at convenient times for participants and covered various times of day, weekdays and weekends. Observation periods were typically around four hours’ duration. Each participant was observed on approximately three occasions.

### 2.2. Ethical Considerations

This study was reviewed and approved by the hospital ethics committee and by the university ethics committee. The researchers emphasised to participants that they were observing usual clinical activities and that there was no expectation for participants to provide explanations or adjust their work priorities to accommodate the researcher presence. If a researcher undertaking observations noted anything that may constitute unsafe medication practice, the individual being observed was to be informed about this issue. Each hospital also had stringent policies and procedures in place to deal with medication errors as they occurred at the ward level.

### 2.3. Data Collection

Semi-structured interviews, observations and focus groups were conducted from April 2018 to October 2019. Semi-structured interviews focused on health professionals’ experiences of communication to manage older patients’ medications across transitions. Observations were used to examine these communication activities in the clinical setting. In reflexive focus groups, health professionals reflected on existing practices and considered opportunities for solution development to address communication problems. Two researchers conducted data collection across both study sites. The researchers undertook reflective journaling following each episode of data collection to enhance understanding. Interview, observation and focus group schedules are presented in Table 1. Schedules were developed from resources provided by the Australian Commission on Safety and Quality in Health Care about communication between health professionals about managing patients’ medications across transitions [22].

### 2.4. Data Analysis

The research team inductively analysed a selection of transcripts using thematic analysis to develop a coding framework, which was applied to the remaining transcripts [24]. Data were coded by two researchers using NVivo 12 Plus (QSR International Pty Ltd., Chadstone, Victoria, Australia) [25]. The research team met fortnightly to discuss findings. Three clinical contexts: transferring setting (preparing for transfer), receiving setting (setting after transfer) and ‘real-time’ (simultaneous communication), were identified as salient during data coding and guided analysis and interpretation of themes and subthemes.

Coding of data excerpts occurred in the following way. Doctors, nurses and pharmacists were given the code Med, RN or Pharm, followed by the number allocated to the particular individual observed. Data collected by interviews did not have a specific code identified, while data from observations or focus groups were assigned the code Obs or FG respectively. In cases where a researcher’s field notes were included, these were designated as FieldNotes. Data collected in the acute hospital were identified by Acute while data from the geriatric rehabilitation hospital were given the code GeriRehab. After the hospital designation, the ward number assigned to a specific ward setting was identified.

## 3. Results

In all, 38 health professionals participated in semi-structured interviews, 29 participated in observations for a total of 203 h and 27 participated in focus groups. Nurse participants were aged between 20–58 years with 2 months to 27 years of clinical experience; pharmacists were aged between 25–62 years with 2 to 40 years of clinical experience; and, doctors were aged 25–40 years with 5 months to 19 years of clinical experience. In all, 58% of doctors, 75% of pharmacists and 88% of nurses who participated were women.

Our findings demonstrated there was limited interprofessional communication to manage medications across transitions of care. As older patients moved from one environment to another, communication tended to be between health professionals of the same discipline. Interprofessional communication between doctors, nurses and pharmacists mostly occurred within a single clinical setting to manage medications after patient admission or when planning for discharge. Health professionals situated in different settings were often unable to access medication information simultaneously, which impacted their ability to identify and investigate potential medication incidents. Most medication communication activities took place either before or after patient transfer. Table 2 summarises themes and subthemes.

### 3.1. Transferring Setting: Interprofessional and Intraprofessional Medication Information Dissemination

Communication in the transferring setting focused on disseminating information to the receiving setting by preparing medication documents and discharge summaries, and through handover conversations. Two subthemes reflected how interprofessional and intraprofessional communication occurred: health professionals’ proactive stance in conveying medication information across communication channels and health professionals’ lack of familiarity with other settings (Figure 1).

#### 3.1.1. Health Professionals’ Proactive Stance in Conveying Medication Information across Communication Channels

Nurses and pharmacists reported that they may experience communication breakdowns when disseminating medication information to receiving settings and that they proactively used both synchronous and asynchronous channels to mitigate this risk. Asynchronous channels of communication included printed discharge medication lists or photocopied hospital medication charts, which were read by receiving health professionals at different times to when these documents were developed. Nurses and pharmacists proactively ‘closed the loop’ on medication information sent asynchronously via fax or physically with the patient (such as discharge medication lists or photocopied hospital medication charts) by also using synchronous channels, often telephone calls, to confirm and clarify medication details, such as changes or unusual dosages, directly with recipients.

During interviews, pharmacists reported how they had learnt from past experiences of ‘fax fails’.


*…we always ring to make sure the faxes have gone through *(to the community pharmacist)*, that the information is there, and that they’re *(community pharmacist)* able to supply what we want. (Pharm5_GeriRehab3)*


Observations confirmed how this process occurred in the clinical setting.

Pharm4:Hello, I am just checking to see if my fax came through.

Pharm4:OK. Thank you. (She waits for them to check).

Pharm4:Wonderful! Now, I haven’t sent the script through yet. Because I just am waiting for the script from the doctors. So, I’ll do for one prazosin, BioMag^®^ (magnesium) and then Nexium^®^ (esomeprazole) 20 s.

Pharm4:OK? Wonderful! Thank you so much. Bye. (Pharm4_Obs_Acute2)

In this situation, the hospital pharmacist was waiting to receive the written medication orders from the hospital doctors before sending through these medication orders to community pharmacists, with any additional instructions relating to the prescribed medications. While hospital doctors wrote the medication orders, it was the hospital pharmacist’s role to distribute this information to community pharmacists.

Similarly, acute care nurses reported how they conveyed the same information through telephone calls, written medication charts and patient notes, in order to reduce the risk of communication breakdowns to the receiving setting. One nurse stated, *“Normally it works, especially if you can jump on the phone and flag that (*medication information*), and I think also sending relevant paperwork helps as well, so it doesn’t get missed.” (RN3_Acute1).*

Some acute care nurses expected that information on photocopied medication charts might not be correctly interpreted by receiving nurses after inter-hospital transfers. This issue prompted nurses to emphasise important information during handover conversations.


*We’d photocopied the medication chart for them *(receiving hospital)* but he’d (older patient) had some diarrhoea for about five days and we’d stopped giving him Coloxyl (docusate sodium) and senna and it was working a treat. I was a bit concerned that when he went there *(other hospital)*, they’d just copy off the medication chart and start giving it to him again because no one looks at a photocopy once he gets back into the hospital. I mentioned that one in the verbal handover to XXX *(receiving hospital)* … “You could just stop it altogether because if it’s on the chart someone’s going to give it to him. He’s not constipated so he doesn’t need it.” Again if I hadn’t given that handover, that wouldn’t have happened. (RN14_Acute3)*


Observations demonstrated how a nurse relayed medication information during a handover conversation to a residential aged care facility and also planned to send a photocopy of the medication chart.


*RN7: We will either fax it to you or either a hard copy. … He is still the same… not much changes. Medications still crushed with puree … He is seen by the speech *(speech pathologist)* here and they put him on level 400. Otherwise, he is OK, saturating well in room air. *(She was being asked some questions by the nursing home staff on the phone)*. Yes, yes. … So, I’ll copy the OBS *(observations)* chart, drug chart, discharge summary. (RN7_Obs_Acute2)*


#### 3.1.2. Health Professionals’ Lack of Familiarity with Unknown Settings

Hospital pharmacists and doctors reported that they were unfamiliar with how community doctors received and understood medication information in the discharge summary. Junior doctors were responsible for writing discharge summaries and hospital pharmacists completed the medication information section of these summaries. However, junior doctors and pharmacists were often uncertain about how or whether the document actually reached the community doctor.


*I try and include everything in my discharge summary, but whether or not the GP *(general practitioner)* is actually reading that and, I guess, understands fully about why we’ve done stuff. (Med1_Acute1)*



*I feel like it would be good to give the GP or the doctor who is looking after them, maybe a bit more information. But I think what happens is the discharge summary is faxed to the GP or someone, but I’m not entirely sure. (Pharm1_Acute1)*


Doctors and pharmacists assumed that the discharge summary reached the community doctor, *“we’re under the assumption that a patient’s discharge summary will go to the GP clinic” (Pharm3_Acute2)*, but did not routinely engage in two-way communication with community doctors to confirm that the summary had been received.


*“If there’s something you really want checked on yeah, you call the GP and be like, “We’re looking at this, this is the story that’s happened,” if there’s something that you really don’t want to go wrong, you call the GP, but not every patient. So, that kind of hand over, which, I mean, maybe they should, but there’s maybe not time.” (Med3_GeriRehab3)*


### 3.2. Receiving Setting: Interprofessional and Intraprofessional Communication for Safe Continuation of Medications

In the receiving setting, interprofessional and intraprofessional communication focussed on the safe continuation of medications from the transferring setting. This focus was reflected in two subthemes: surveillance and investigation of medication incidents, and health professionals’ commitment to continuity of medications across transitions of care (Figure 1).

#### 3.2.1. Surveillance and Investigation of Medication Incidents

Nurses and pharmacists attempted to identify medication incidents as older patients transferred from other environments to their care. This identification included rechecking of medication appropriateness or unclear orders, *“someone was recently admitted, found to have either ischaemic or haemorrhagic stroke, and then they’re charted Clexane^®^* (enoxaparin). *I’ll always get it reviewed by the doctor just to confirm that they actually do want to give that.” (RN2_Acute1)*

When admitting patients, nurses studied medication charts for incidents, such as medications that were due to be given, but were missing a signature to indicate that they were administered. Nurses were aware of potential safety risks to older patients if medications were inadvertently double-dosed during a transfer.


*The medication chart—it was from another ward—their midday dose, I think it was of their frusemide (furosemide) wasn’t given, well it wasn’t signed for, so we followed that exact process. Couldn’t chase up the nurse, because by the time we found it in the transfer it was the AM, and it was an agency nurse or something; it was not going to be followed up any time soon, anyway. So we did page the team. They came to review the patient clinically, ‘yes, looks overloaded, give the dose, it’s fine if we give extra’. (RN5_Acute1)*


*‘Chasing up’* referred to how nurses in receiving settings attempted to communicate back to the nurses at the transferring setting, to discuss potential medication incidents identified after transfer. Receiving nurses reported they were sometimes unable to clarify medication issues because transferring nurses were not available due to shift changes or due to difficulties identifying the correct nurse.


*If they have come from another facility, it can be really hard to fix that. … If they have come from another hospital in a different city, out of state, out of country, then it’s much harder to call, because you have got to call the hospital, hopefully get put through to the ward that they were on, but because they are discharged from their hospital now, the switchboard can’t necessarily direct you to where they were, so you can’t necessarily find the right nurse to ask, it might not be their shift, they may not have any idea whether that medication was given. The doctors won’t know, because they don’t give the medication so chasing it up is really, really hard. (RN10_Acute2)*


Observations demonstrated how a nurse in the acute setting identified a documentation incident after patient transfer. The nurse telephoned back to the emergency nurse to resolve the incident.

RN6 (on phone to emergency nurse):It’s just that her NGT (nasogastric tube) is in, but there’s no proper documentation so we can’t use it *[for medication administration]*. Do you know who put the NGT in?

RN6:Yeah so we can’t really use that in the ward. So it needs to be properly documented on the NGT form. By the nurse who inserted it, at what centimetre *[for positioning]* and all that stuff.

RN6:We do have the forms here, if you could get the nurse who looked after her just to come up and fill out the forms, is that alright? (RN6_Obs_Acute2)

The documentation incident and the need to ‘chase up’ with the emergency nurse caused delays to medication administration as reported by the nurse the following day, “*I was able to chase it up, but it was a little bit late*.” *(RN6_Acute2)*

After patient transfers, pharmacists examined discharge summaries, paper and electronic medication charts and the patients’ own medications for prescribing and dispensing incidents. The following observation excerpt captures this meticulous surveillance:


*The pharmacist is sitting at a satellite station on the ward at the computer. She is reviewing the patient’s medication reconciliation form *(MRF)*. (…) The pharmacist [then] stands up and opens the patient’s locked medication cupboard at the back of the satellite station. She takes out the patient’s medication bucket, which contains the patient’s regular medications in a red bag *(hospital supplied to store patient’s own medications)*. The pharmacist opens the bag and looks at the medication containers and bottles. She opens some of the containers and appears to look at how many tablets are inside. The pharmacist replaces the medications and closes the cupboard. She collects the patient’s medication chart and walks to the next satellite station to find the intern *[to discuss a medication issue identified].* (Pharm1_Acute1_FieldNotes)*


Hospital pharmacists reported how they identified medication incidents that had sometimes led to older patients’ hospital admission.

*So the medication was rivaroxaban, it’s used for AF* (atrial fibrillation)—*prevention of stroke. This patient came in with a stroke. So potentially, we could have prevented the stroke if the dose was appropriate. So apparently, it had been adjusted for their renal function in the past, which had been poor, but now their renal function was pretty much fine and they could have had their full dose. (Pharm1_Acute1)*


*So the one I’ve had is that the pack said that it should be metoprolol 50 mg BD, but what was being packed was actually 100 mg BD (two times daily), so it was double the dose. So then looking at first of all is the patient’s safety kind of thing, is that why the patient’s come into hospital and what can we do to fix that? And in this case, she was bradycardic and it potentially could have been why she was admitted. (Pharm2_Acute2)*


Observations demonstrated how a geriatric rehabilitation pharmacist identified a medication prescribing incident when she reviewed the online medication record after the patient transferred from the acute hospital.


*Ahh, no that [prednisolone] shouldn’t be seven days. That’s wrong, so we don’t [want] them to stop it in seven days. No. No duration, thank you! (Pharm5_Obs_GeriRehab2)*


#### 3.2.2. Continuity of Medications from Transferring Settings

Doctors, nurses and pharmacists in all settings facilitated the continuation of older patients’ medications as they moved between environments. Communication between nurses in transferring and receiving settings sought to manage the supply of medications across transfers to prevent delayed or missed administration. Geriatric rehabilitation settings had less access to pharmacy services and medication stock, particularly after hours. In these settings, nurses who were waiting for patients’ arrival reported that they asked nurses in transferring acute care settings to send certain medications with patients to ensure sufficient availability of stock to administer medications on time.


*…when patients transfer, what we always do is we call them for a handover and we always say, “Could you please put in all the non-imprest *(non-ward stocked)* medication when they come here.” Because we’re not a major hospital … the pharmacy department don’t open 24/7. They probably close roughly about 5:30 so, we’ve got to make sure that all of that comes over with [the patient so]… on arrival to us, that it’s there. (RN20_GeriRehab4)*


In all hospital settings, doctors acknowledged the pharmacists’ role in medication reconciliation across patient transfers, and valued their ability to source accurate information.


*“We’re very lucky here that our pharmacists do a lot … for us so they will liaise with the actual chemist in the community to see what’s being dispensed to the patient and they will check the patient’s physical medications if they’ve got them” (Med2_FG_Acute).*


There were competing priorities, such as time pressures, that impacted delayed and missed prescribing of vitamins and eye drops across transfers. Pharmacists and nurses reported that doctors perceived there was less urgency to prescribe *‘simple’* medications, such as vitamins and eye drops across transfers, even though pharmacists and nurses considered these medications to be important to older patients.


*So with the medical staff being so time-poor, they would chart whatever they felt were the most necessary medications at the time and not chart anything else, whereas as part of our credentialing process, we have to think about all the medications. … But even still, I would feel that the patients sort of arrive up on the ward and they’d still have a handful of medications still not charted. So it’s up to us to sort of follow-up on that and say ‘I know, it’s just magnesium *(laugh)* or Vitamin D. But for consistency, we need to carry on with these’. (Pharm3_Acute2)*



*We’ve had a few times where things *(medications)* have been left for a really long time. It’s simple things like eye drops and things that are really important to them (older patients) and things like that. We’re not getting them ordered and things so I just feel like we take a really more laid back approach. (RN11_Acute2)*


During observations, a medication incident occurred in the geriatric rehabilitation setting where an older patient informed the bedside nurse that his regular lubricating ‘Refresh’ eye drops had not been prescribed since his admission. The nurse forgot to notify the doctors initially but remembered later in her shift.


*RN17 (to nurse in charge): Ah! I forgot to tell the doctors. I’ll tell them to order him up some normal [eye] lubrication. I don’t think he’s got conjunctivitis. He’s just got the upturned eyelids, it looks irritated and dry. (RN17_Obs_GeriRehab4)*


This failure to prescribe the patient’s regular eye drops across transitions caused harm as he experienced discomfort in his eyes.

### 3.3. Real Time Communication: Barriers to Collaborative Interprofessional and Intraprofessional Communication

Communication practices impacted collaboration that occurred in the moments when health professionals connected in real time to manage medications across transitions of care. Communication during these interactions was reflected in two subthemes: discontinuous nursing handover responsibilities and missed opportunities for inclusive communication to manage medications (Figure 1).

#### 3.3.1. Discontinuous Nursing Handover Responsibilities

Responsibility for handover conversations was discontinuous as it moved between different nurses within and across shifts, as they covered break times and shared tasks to accommodate competing priorities. Nurses were sometimes required to hand over details about patients about whom they were not familiar. One acute care nurse recounted during an interview how this situation could affect the accuracy of medication information communicated during handover.


*I guess if I’m not really familiar with that patient so if I’ve just come on to the shift and then I get handed the phone to hand over this patient that I’ve only just met and things. I guess I can look at the medication chart and things but if they *(receiving nurse)* ask specific questions then the answer might not 100% be correct. I feel sometimes when we ask that from other places as well you get the same sort of thing. A lot of time you say, “I’ve only just got handover on them. I’ve only just met them.” So I feel like that’s a really big barrier in some ways. (RN11_Acute2)*


In the following observation excerpt, the oncoming afternoon nurse had just received handover of an older patient from the morning nurse, who was on break when the patient transport officer arrived to transfer the patient.

The afternoon nurse:So, I’ve only just started my shift by the way.

The ambulance attendant:Ok

The afternoon nurse:So I’ve barely laid eyes on her (older patient). So she had this left intraparenchymal haemorrhage, which I’m not really that familiar with to be honest. She has got a history of AF (atrial fibrillation) there. Which I think she is on metoprolol for… (RN3_Obs_Acute1)

The phrases, *‘I’ve only just started my shift’* and *‘I’ve barely laid eyes on her’* signal the nurse’s unfamiliarity with the patient to the patient transport officer.

#### 3.3.2. Missed Opportunities for Inclusive Communication to Manage Medications

There were limited opportunities for pharmacists to be involved in face-to-face discussions with doctors, nurses and other pharmacists about medication management and transfer planning. Nurses were constantly present within the ward environment, junior doctors less so and senior doctors generally only participated in scheduled ward rounds. Pharmacists were assigned to patients according to their medical specialty, which meant they were sometimes required to visit patients across different wards.

Pharmacists often had to seek out information themselves as they were not routinely or proactively informed about changing plans, which left them feeling ‘*not kept in the loop’ (Pharm2_Acute2)*. Hospital pharmacists reported they might support inter-ward transfers with a verbal handover, but this was not routine practice.


*I would say occasionally you might get the other pharmacist from the other team giving you a handover especially if there’s something to follow up. But that doesn’t happen all that often. Especially if it’s a surgical patient, they’re just like turfed to XXX *(medical unit)* and you might find out later on, ‘oh I’ve got a transfer of care’. You might be looking at your medication list on *[electronic system]* and then all of sudden there’s someone new and you think ‘Ugh! Where did they come from?!’ A lot of the pharmacists are in their medical teams WhatsApp groups, and so they’ll get notification often from that perspective, like take over of care from… (Pharm1_FG_Acute1)*


The phrase, *‘all of a sudden there’s someone new…’* reflected the surprise felt by pharmacists upon the unexpected discovery of a new patient admission. The WhatsApp groups were originally set up for doctors to converse with each other about changes in patient transfers. Pharmacists perceived that it was a privilege to be included in these groups. While this communication channel enabled notification of impending transfers, it also added to the complexity of the variety of ways in which pharmacists could be notified about new admissions. Observations demonstrated how pharmacists received delayed updates from doctors and nurses.

Pharm5 to doctor:Hi. He (older patient) is alright. He is not happy that he is not going home today. But I just said that the doctor will come and tell him that it is definitely tomorrow. I think if he gets that it’s definitely tomorrow then he’ll probably be ok.

Doctor:It depends; he can go to home today to be honest.

Pharm5:Really?

Doctor:If Endo (endocrinology team) are happy with him.

Pharm5:Oh. Ok. But the daughter?

Doctor:The daughter is happy for him-

Pharm5:Today?

Doctor:Yeah, yeah, yeah.

Pharm5:Oh.

Doctor:She’s happy for him to go home today. She was happy for him to going yesterday.

Pharm5:Oh, ok. Well then I should-

Doctor:[Today or tomorrow]

Pharm5:Alright. So, we are just waiting for Endo then..

Doctor:Yeah.

Pharm5:I’ll get some meds ready.

Doctor:Yeah. Well-yeah.

Pharm5:Well, I don’t know the- yeah. Ok. (Pharm5 to herself): (sighs) You wonder why patients get annoyed, I don’t know even know what’s going on. It says tomorrow. … Well (sighs)… anyway. … I’m going to get some scripts ready. (Pharm5_Obs_GeriRehab2)


*Pharmacist asks question to the nurse what happen to him *(older patient)* and where he went. Nurse tells that the older patient was already discharged and was picked up by transfer staff minutes ago. Pharmacist becomes very upset and frustrated as she missed the opportunity to explain to the older patient about his medications. She decides that she will call the rehabilitation hospital where the patient was discharged *(and contact pharmacists working there)* and gives a handover to them over the phone. (Pharm4_Acute2_FieldNotes)*


These excerpts demonstrate how delayed interprofessional communication about changes to transfer plans meant that pharmacists had to reorganise their medication activities.

## 4. Discussion

The aim of the paper is to examine how interprofessional and intraprofessional communication occurs in managing older patients’ medications across transitions of care in acute and geriatric rehabilitation settings. The findings provided insights into complex patterns of interactions between health professionals in managing medications for older patients across transitions of care. As patients moved across transitions of care, health professionals at the transferring settings disseminated medication information, while health professionals in the receiving settings confirmed the safe continuation of prescribed medications. Across transferring and receiving settings, barriers to effective collaboration were apparent in interactions between health professionals. Communication surrounding transfers often tracked back and forth between health professionals across various communication channels as they disseminated information before transfers and attempted to investigate medication incidents after transfers. Health professionals of different disciplines rarely communicated with each other between settings; rather, communication occurred between health professionals of the same discipline. Communication processes were often unreliable, which led to nurses and pharmacists proactively initiating additional communication activities to prevent potential communication breakdowns. There were unclear processes for disseminating discharge summaries and transfer of accountability to community doctors. Pharmacists were not routinely informed about changing plans, despite their important role in managing medications at transition points.

Unreliable processes for transferring medication information impacted how nurses and pharmacists communicated across settings at transition points. They perceived information transmission via paper documents and fax machines to be particularly unreliable due to the potential for important information to be missed and their past experiences of fax machines failing. They often compensated for these unreliable processes by instigating additional activities to minimise the risk of communication breakdowns and to close communication loops. Asynchronous channels meant that there was a time lag between when the document was prepared by the transferring setting to when it was read by the receiving setting. Although previous reports have described that asynchronous communication channels can be less interruptive than synchronous communication, which requires an individual to respond immediately [26,27], our findings showed that during transfers, communication via paper documents or fax was often supplemented with phone calls, which in fact caused interruptions.

Interestingly, these risk minimisation strategies were not routinely performed when disseminating discharge information to the community doctor. This situation was influenced by unclear communication systems for how discharge summaries were distributed and by unclear transfer of accountability from hospital to community doctors. Previous studies have reported the discontinuity of care between hospital and community doctors due to inadequate or delayed transfer of discharge information [28,29]. Our study demonstrated how lack of familiarity about how community doctors received discharge information may have contributed to communication breakdowns between the two settings. It was also uncertain how and when transfer of accountability moved from the hospital doctor to the community doctor at discharge. Our findings suggested that transfer of accountability predominantly occurred asynchronously via the discharge summary, which was created at a different time to when the community doctor received the information. Previous work has also shown that transfer of accountability and transfer of information can be disconnected and unclear at discharge, which could impact safe and accurate continuation of medications [30,31]. Greater understanding and attention to communication are needed between these settings to determine preferences and effective measures for communication at discharge.

There were patterns of inter-nurse communication across settings that compensated for challenges related to discontinuous clinical roles and responsibilities. The act of ‘chasing up’ medication discrepancies after transfers was routine, but was hindered by difficulties in accessing the appropriate nurse due to changing shifts and responsibilities for patient care, which has been seen in past work [32]. Nurses in transferring settings also experienced challenges when communicating clinical handover if they were not familiar with the patient. Similar to other work, the unpredictable timing of patient transfers and nature of the clinical setting impacted how handover between settings occurred [33,34], which at times necessitated the need for another nurse to step in and cover for an occupied nurse. For instance, in a past nursing group interview study [33], strained relationships were often identified during handover. In that study, while nurses in transferring and receiving wards understood that certain information was desirable, sometimes they did not share or prioritise this information. Rather, nurses commented about contextual pressures that contributed to their inability to meet the receiving nurses’ information needs. As demonstrated in the current study, nurses recognised that pressures faced by their nursing colleagues meant certain information could not be delivered comprehensively and in a timely way. Our findings also demonstrated that the ward processes for covering break times led to nurses who were not the primary nurse or who had just started their shift, being responsible for communicating handover. This study showed how nurses accepted and fulfilled this duty but used language that signalled their unfamiliarity of the patient to the handover recipient. Effective clinical handover is important for reducing errors across transitions [35] and improvements are needed to reduce nurses’ reliance on fall-back communication measures.

Missed opportunities for inclusive communication with pharmacists restricted shared expertise in collaborating about managing medications across transitions. Pharmacists experienced delayed notification of transfer information and were not always routinely updated about changing medication or transfer plans. This was surprising, as doctors and nurses acknowledged and valued the role of pharmacists to manage medications at admission and discharge; yet, the processes for updating pharmacists about changing plans at these transition points were unreliable. Our findings support earlier work, which found that patients were sometimes transferred without pharmacists’ knowledge [12], and a lack of intraprofessional pharmacist handover between ward transfers was also seen in a study examining patient transfers between emergency departments and medical wards [19]. Another study similarly reported missed opportunities of interprofessional collaboration to enhance medication safety and frequent impromptu interactions [36]. Our findings demonstrated that impromptu updates to pharmacists often conveyed delayed information and impacted timely medication management. Pharmacists did not seem to challenge the status quo, and accepted and expected delayed updates from other health professionals (albeit with frustration) and then adapted their clinical activities to meet revised plans.

There are a number of implications for practice arising from the study findings in maintaining medication safety at transitions of care. First, there needs to be greater understanding among all health professional disciplines about when pharmacists need to be notified about changing medication or transfer plans. Second, attention to improving clinical handover communication at the time of transfer is needed as greater levels of proactive preparation by health professionals would reduce opportunities for errors relating to continuation of medications. Third, hospitals need to establish clear processes for how health professionals should disseminate discharge summaries to community doctors and transfer accountability to close the loop with medication management as patients move between settings. Fourth, there needs to be greater levels of interactions between different health professional disciplines across settings.

### Limitations

There are some limitations to this study. First, it is possible that researcher presence may have created an observer effect in how participants communicated. However, researchers had a prolonged presence in each setting, which helped to promote familiarity. Participants were reassured that the intent was not to assess or judge their practice. Second, transfer communication often occurred between two settings but researchers could only observe one side of the information exchange. Further research could involve simultaneous observations of transferring and receiving care settings to improve understanding of how medication information is disseminated and received across transitions of care. Third, this study was undertaken in metropolitan hospitals and therefore the findings may not necessarily be transferable to regional and rural areas.

## 5. Conclusions

This paper has identified patterns of interprofessional and intraprofessional interaction as communication occurred within and across transferring and receiving settings to manage medications for older patients. Communication processes across transferring and receiving settings were unreliable and health professionals instigated additional routine measures to minimise the risk of communication breakdowns. Medication safety was compromised across transitions of care due to unclear processes for disseminating discharge information and transfer of accountability to community doctors. Pharmacists often received delayed notifications of changing medication and transfer plans, despite all health professional disciplines acknowledging their significant role to manage medications at admission and discharge.

## Figures and Tables

**Figure 1 ijerph-18-03925-f001:**
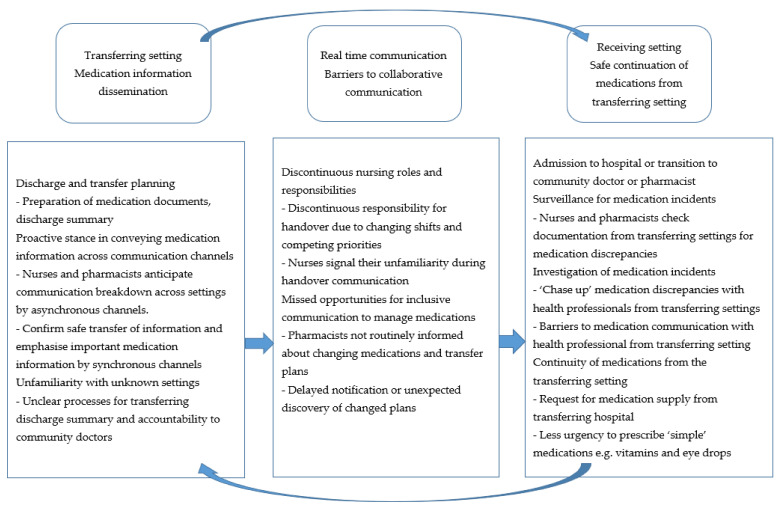
Interprofessional and intraprofessional medication communication between the transferring setting, receiving setting and real time communication.

**Table 1 ijerph-18-03925-t001:** Interview, observations and focus group schedules.

Semi-Structured Health Professional Interview Schedule
Examples of interview questions: What helps you to communicate effectively about older patients’ medications when they move from one place to another? Can you talk about a particular experience? What do you say to other health professionals situated in the community about medications as older people move from your ward to their homes? What do you say to health professionals situated in a residential aged care facility about medications as older people move from your ward to that care facility? What examples can you provide from your own clinical experience of how you successfully managed communication problems involving medications as older people move across settings?
Observations schedule
Describe what the health professional is saying about the medications. State the location of where communication occurs and who is present. Describe the communication strategies used during the communication encounter.
Focus Group interviews
Quotation excerpts of communication practices and interactions from interviews and observations were read to participants. Participants were asked to reflect on these existing practices and to consider possible strategies for improvement [23].

**Table 2 ijerph-18-03925-t002:** Summary of themes and subthemes.

Themes	Transferring Setting: Medication Information Dissemination	‘Real-Time’: Barriers to Collaborative Communication	Receiving Setting: Safe Continuation of Medications from Transferring Setting
**Sub Themes**	1. Proactive stance in conveying medication information across communication channels 2. Unfamiliarity with unknown settings	1. Discontinuous nursing roles and responsibilities 2. Missed opportunities for inclusive communication to manage medications	1. Surveillance and investigation of medication incidents 2. Continuity of medications from transferring settings

## Data Availability

Further details regarding where data supporting reported results can be found, are to be directed to the first author.

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
