# Peer review of "Interprofessional and Intraprofessional Communication about Older People’s Medications across Transitions of Care"

_ijerph, 2021, doi:10.3390/ijerph18083925_

Round 1

Reviewer 1 Report

Dear Authors

Many thanks for the opportunity to review your article, which eloquently describes your investigation into interprofessional and intraprofessional communication about medicines across transitions of care. I have the following (minor) observations, which are intended to provide further clarity for the international reader, looking at your work for the first time.

Abstract - This is very well written, and I suspect this was constrained by a word count. I do think there is potential to further highlight some of the important points in the discussion and conclusion regarding interprofessional and intraprofessional communication. I think a key message, reflecting the title, is that health professionals of different disciplines rarely communicated with each other, rather the communication occurred between health professionals of the same discipline. Also the last line of the conclusion might also be reflected in the abstract more - that lack of communication with pharmacists occurred despite all healthcare professionals acknowledging their significant role.

Introduction. I think it might be helpful to consider a paragraph outlining the intended "normal" practice for communicating when patients are transferred. I am not sure if fax is the normal procedure, or perhaps it is in this context. This would help understand the data better in the results section.

Section 2.2 Ethical Considerations. I am sure that you were obligated to have a procedure to ensure safety netting is an issue was identified during observations. It may be helpful to outline this procedure and also the procedure re verbal consent.

P6 L182-183. I am not sure what this means, and again might be helpful to explain the normal practice. I'm not sure what they are going to "do one for" as the procedure appears to rely on a script from a doctor.

P6 L197. Can you insert docusate (I believe) for the international reader?

P7L259. Can you insert furosemide for the international reader?

P10L361. Note who it is that considers eye drops etc as important to older patients.

P12L429. This is another point that might be helpful in the introduction to provide context. Pharmacists being assigned to medical specialty rather than specific wards.

P12L436. I suggest review of this quote. It raises some separate issues re language and the method of communication between teams and the pharmacists. The key phrase is "all of a sudden there's someone new" so you might consider abbreviating to capture that.

P12 last line and to P13. I wasn't entirely clear if this conversation between a doctor and a pharmacist was a moot point as the patient had already been discharged. Was that the intention to convey, and if so you might make that more explicit?

Limitations. There is no issue with the limitations that you raise, but I would encourage you to think about including patients and carers in future research, as I believe their perspective is important.

Wishing you all the very best with your research

Author Response

Interprofessional and intraprofessional communication about older people’s medications across transitions of care, manuscript no. IJERPH-1154400

 Response to Reviewer 1

 Open Review

English language and style

( ) Extensive editing of English language and style required
( ) Moderate English changes required
(x) English language and style are fine/minor spell check required
( ) I don't feel qualified to judge about the English language and style

Yes

Can be improved

Must be improved

Not applicable

Does the introduction provide sufficient background and include all relevant references?

(x)

( )

( )

( )

Is the research design appropriate?

(x)

( )

( )

( )

Are the methods adequately described?

(x)

( )

( )

( )

Are the results clearly presented?

(x)

( )

( )

( )

Are the conclusions supported by the results?

(x)

( )

( )

( )

Comments and Suggestions for Authors

Dear Authors

Many thanks for the opportunity to review your article, which eloquently describes your investigation into interprofessional and intraprofessional communication about medicines across transitions of care. I have the following (minor) observations, which are intended to provide further clarity for the international reader, looking at your work for the first time.

Abstract - This is very well written, and I suspect this was constrained by a word count. I do think there is potential to further highlight some of the important points in the discussion and conclusion regarding interprofessional and intraprofessional communication. I think a key message, reflecting the title, is that health professionals of different disciplines rarely communicated with each other, rather the communication occurred between health professionals of the same discipline. Also the last line of the conclusion might also be reflected in the abstract more - that lack of communication with pharmacists occurred despite all healthcare professionals acknowledging their significant role.

Response: The following text has been added to the abstract (p. 1, lines 29-32):

Although health professionals of different disciplines sometimes communicated with each other, communication mostly occurred between health professionals of the same discipline. Lack of communication with pharmacists occurred despite all health professionals acknowledging their important role.

Introduction. I think it might be helpful to consider a paragraph outlining the intended "normal" practice for communicating when patients are transferred. I am not sure if fax is the normal procedure, or perhaps it is in this context. This would help understand the data better in the results section.

Response: The following paragraph has been added in the Introduction to explain the usual practice undertaken by health professionals when communicating about medications when patients are transferred (p. 1, line 45 – p. 2, line 64):

There is a ‘usual practice’ for communicating across different transitions of care during the journey between home or aged care facilities, admission to hospital, transfers within hospital, discharge home or to aged care facilities, and movements within community care facilities [2]. In the emergency department, prior to admission, a medication history is taken either by a pharmacist, or by an emergency doctor. The admitting doctor examines the medication history and medical record of the individual, including checking any previous admission to the hospital. Nurses complete a verbal handover between the emergency department and admitting ward. Discussions may also occur between the admitting doctor and senior consultant of the admitting ward or more junior medical staff may be involved. Pharmacists in the admitting ward are notified about a patient’s medication requirements by the admitting doctor’s written notes in the medical record and their notations in the medication chart. However, pharmacists tend to be assigned to medical specialties rather than specific wards. In communicating about a patient’s movements within a hospital, verbal handovers are usually conducted individually with single discipline groups comprising nurses, doctors, or pharmacists. Following discharge from hospital, a discharge summary is written, with information about the medications to be prescribed. Hospital doctors relay this discharge summary to the general practitioner or primary care doctor using secure electronic messaging systems or written letters. Facsimile use to convey information at the time of hospital discharge remains common in countries such as the United States, Australia, and the United Kingdom [5,6].

Section 2.2 Ethical Considerations. I am sure that you were obligated to have a procedure to ensure safety netting is an issue was identified during observations. It may be helpful to outline this procedure and also the procedure re verbal consent.

Response: The following text has been added in detailing what was to occur if an unsafe practice was observed (p. 3, lines 121-122):

If a researcher undertaking observations noted anything that may constitute unsafe medication practice, the individual being observed was to be informed about this issue.

The following text has been added to clarify the process of verbal consent with health professionals who were not the formally consented individuals being observed (p. 3, lines 106-112):

With regards to other health professionals interacting with the formally consented participant, verbal consent was sought during information sessions before the conduct of observations. Alternatively, verbal consent was sought during the time of the observations, if those individuals were not available to attend information sessions. If a specific health professional chose not to have their conversations noted during observations, any recording of information involving that person immediately ceased.

P6 L182-183. I am not sure what this means, and again might be helpful to explain the normal practice. I'm not sure what they are going to "do one for" as the procedure appears to rely on a script from a doctor.

Response: The following text has been added (p. 7, lines 222-226):

In this situation, the hospital pharmacist was waiting to receive the written medication orders from the hospital doctors before sending through these medication orders to community pharmacists, with any additional instructions relating to the prescribed medications. While hospital doctors wrote the medication orders, it was the hospital pharmacist’s role to distribute this information to community pharmacists.

P6 L197. Can you insert docusate (I believe) for the international reader?

Response: The text has been changed to the following (p. 8, line 239):

Coloxyl (docusate sodium)

P7L259. Can you insert furosemide for the international reader?

Response: The text has been changed to the following (p. 9, line 302):

I think it was of their frusemide (furosemide)

P10L361. Note who it is that considers eye drops etc as important to older patients.

Response: The text has been changed to the following (p. 12, lines 401-404):

Pharmacists and nurses reported that doctors perceived there was less urgency to prescribe ‘simple’ medications, such as vitamins and eye drops across transfers, even though pharmacists and nurses considered these medications to be important to older patients.

P12L429. This is another point that might be helpful in the introduction to provide context. Pharmacists being assigned to medical specialty rather than specific wards.

The following text has been added in the Introduction. This addition is part of a comprehensive explanation providing details about the context in which health professionals communicate with each other in managing patients’ medications across transitions of care (p. 2, lines 56-57):

Response: However, pharmacists tend to be assigned to medical specialties rather than specific wards.

P12L436. I suggest review of this quote. It raises some separate issues re language and the method of communication between teams and the pharmacists. The key phrase is "all of a sudden there's someone new" so you might consider abbreviating to capture that.

Response: The author team has decided to leave the key phrase “all of a sudden there’s someone new” within the quote because it conveys the spontaneous and unstructured way in which pharmacists were notified about a new admission. The following text has been added to clarify the communication channel relating to the use of the WhatsApp group (p. 14, lines 487-492):

The WhatsApp groups were originally set up for doctors to converse with each other about patient transfers. Pharmacists considered it was a privilege to be included in these groups. While this communication channel enabled notification of impending transfers, it also added to the complexity of the variety of ways in which pharmacists could be notified about new admissions.

P12 last line and to P13. I wasn't entirely clear if this conversation between a doctor and a pharmacist was a moot point as the patient had already been discharged. Was that the intention to convey, and if so you might make that more explicit?

Response: The excerpt mentioned on p. 14, lines 495-522 is different to the excerpt mentioned on p. 14, lines 518-523. These are two different excerpts, comprising two different situations in different wards. The first excerpt relates to a patient being discharged home, and the second excerpt relates to a patient being transferred to a rehabilitation hospital. These excerpts have different codes assigned to them as indicated by the text in brackets. From the reviewer’s comment, it appears as though the assumption was made that these two excerpts were in fact only one excerpt.

Limitations. There is no issue with the limitations that you raise, but I would encourage you to think about including patients and carers in future research, as I believe their perspective is important.

Response: We agree that it is important to examine findings pertaining to patients and their families. These findings will be reported in future publications.

Wishing you all the very best with your research

Submission Date

06 March 2021

Date of this review

19 Mar 2021 23:07:44

Reviewer 2 Report

First of all, this is an interesting issue that the author use semi-structured interview to discover the communication of older people’s medications across transitions of care.

According the journal’s delination, the abstract should be a total of about 200 words maximum.

Introduction

The author lacks a definition of whether there is a routine process for across transitions in Australia or other countries.

In line 51, the sequence of reference is usually ordered by number in the same sentence, it should be [3,5].

Methods

         The code sof subjects should to be exame here.

Results

         The figure 1 should replaced as a high- quality one and add footnote of the meaning

Dissculssion

In the first paragraph, there should to exame the outline of subject in this study then to mention the main outcome.

In the subsequent paragraphs, it is recommended to divide it several summaries in accordance with the results for the readers convenience.

The thinking unit obtained by the interview is not sufficiently discussed with related literature

Author Response

Interprofessional and intraprofessional communication about older people’s medications across transitions of care, manuscript no. IJERPH-1154400

Reviewer 2

Review Report Form

Open Review

English language and style

( ) Extensive editing of English language and style required
(x) Moderate English changes required
( ) English language and style are fine/minor spell check required
( ) I don't feel qualified to judge about the English language and style

Yes

Can be improved

Must be improved

Not applicable

Does the introduction provide sufficient background and include all relevant references?

( )

( )

(x)

( )

Is the research design appropriate?

(x)

( )

( )

( )

Are the methods adequately described?

( )

(x)

( )

( )

Are the results clearly presented?

(x)

( )

( )

( )

Are the conclusions supported by the results?

( )

( )

(x)

( )

Comments and Suggestions for Authors

First of all, this is an interesting issue that the author use semi-structured interview to discover the communication of older people’s medications across transitions of care.

Response: We wish to clarify that we used observations, semi-structured interviews and focus groups as data collection methods in our study.

According the journal’s delination, the abstract should be a total of about 200 words maximum. [sic]

Response: The abstract has now been reduced to a total of 199 words (p. 1, lines 17-33).

Introduction

The author lacks a definition of whether there is a routine process for across transitions in Australia or other countries. [sic]

Response: The following text has been added in the Introduction to provide details about usual practice that is followed across transitions of care (p. 1, line 45 – p. 2, line 64):

There is a ‘usual practice’ for communicating across different transitions of care during the journey between home or aged care facilities, admission to hospital, transfers within hospital, discharge home or to aged care facilities, and movements within community care facilities [2]. In the emergency department, prior to admission, a medication history is taken either by a pharmacist, or by an emergency doctor. The admitting doctor examines the medication history and medical record of the individual, including checking any previous admission to the hospital. Nurses complete a verbal handover between the emergency department and admitting ward. Discussions may also occur between the admitting doctor and senior consultant of the admitting ward or more junior medical staff may be involved. Pharmacists in the admitting ward are notified about a patient’s medication requirements by the admitting doctor’s written notes in the medical record and their notations in the medication chart. However, pharmacists tend to be assigned to medical specialties rather than specific wards. In communicating about a patient’s movements within a hospital, verbal handovers are usually conducted individually with single discipline groups comprising nurses, doctors, or pharmacists. Following discharge from hospital, a discharge summary is written, with information about the medications to be prescribed. Hospital doctors relay this discharge summary to the general practitioner or primary care doctor using secure electronic messaging systems or written letters. Facsimile use to convey information at the time of hospital discharge remains common in countries such as the United States, Australia, and the United Kingdom [5,6].

The following paragraph has also been included to provide information about the Standard Operating Protocol, developed by the World Health Organization to support health professionals globally about strategies to follow when managing patients’ medications across transitions of care (p. 2, lines 65-70):

The World Health Organization (WHO) developed the International High 5s Project to address issues involving patient safety. In regards to medication accuracy at transitions of care, the WHO also created a Standard Operating Protocol to provide health professionals globally with strategies to enhance communication about their medication activities [7]. The goal of this initiative was to reduce undocumented medication discrepancies and prevent medication-related harm as patients moved between settings.

In line 51, the sequence of reference is usually ordered by number in the same sentence, it should be [3,5].

Response: The sequence of citations has been ordered by increasing number in the same sentence. The citation numbers have changed because additional references have been added to the Introduction. The sentence now reads as (p. 2, lines 71-72):

Recent research examining medication management across transitions of care has focussed on medication incidents [3,8]…

Methods

The code sof subjects should to be exame here. [sic]

Response: The following explanation has been given about the coding used (p. 4, lines 150-157):

Coding of data excerpts occurred in the following way. Doctors, nurses and pharmacists were given the code Med, RN or Pharm, followed by the number allocated to the particular individual observed. Data collected by interviews did not have a specific code identified, while data from  observations or focus groups were assigned the code Obs or FG respectively. In cases where a researcher’s field notes were included, these were designated as FieldNotes. Data collected in the acute hospital were identified by Acute while data from the geriatric rehabilitation hospital were given the code GeriRehab. After the hospital designation, the ward number assigned to a specific ward setting was identified.

Results

The figure 1 should replaced as a high- quality one and add footnote of the meaning

Response: We have replaced the figure with a higher quality one, and have positioned the figure in a landscape position on its own (p. 6). The header of the figure has been changed to clarify its meaning and intent.

Dissculssion

In the first paragraph, there should to exame the outline of subject in this study then to mention the main outcome. [sic]

Response: We included the outline of the main aim of the study, and the key themes (outcomes) that were found (p. 15, lines 528-536):

The aim of the paper is to examine how interprofessional and intraprofessional communication occurs in managing older patients’ medications across transitions of care in acute and geriatric rehabilitation settings. The findings provided insights into complex patterns of interactions between health professionals in managing medications for older patients across transitions of care. As patients moved across transitions of care, health professionals at the transferring settings disseminated medication information, while health professionals in the receiving settings confirmed the safe continuation of medications prescribed. Across transferring and receiving settings, barriers to effective collaboration were apparent in interactions between health professionals.

In the subsequent paragraphs, it is recommended to divide it several summaries in accordance with the results for the readers convenience.

Response: In the discussion, we provided four comprehensive paragraphs where we interpreted and explained how the three main themes interacted with each other. If we present a summary of each theme and subtheme in the discussion, this would be a linear depiction of practices and communication taking place, which would not accurately portray the complexity of what occurred. Table 2 already provides a summary of the themes and subthemes analysed in the study. In addition, Reviewer 1 was satisfied with the presentation of the Discussion.

The thinking unit obtained by the interview is not sufficiently discussed with related literature

Response: We have discussed our interview results relating to the interactions that existed during nursing handovers with findings from previous relevant research (p. 16, lines 587-595):

For instance, in a past nursing group interview study [33], strained relationships were often identified during handover. In that study, while nurses in transferring and receiving wards understood that certain information was desirable, sometimes they did not share or prioritise this information. Rather, nurses commented about contextual pressures that contributed to their inability to meet the receiving nurses’ information needs. As demonstrated in the current study, nurses recognised that pressures faced by their nursing colleagues meant certain information could not be delivered comprehensively and in a timely way.

Response: We have carefully read the whole manuscript, in order to check for English language and style.

Submission Date

06 March 2021

Date of this review

18 Mar 2021 09:08:32

Bottom of Form

© 1996-2021 MDPI (Basel, Switzerland) unless otherwise stated

Round 2

Reviewer 2 Report

This revised manuscript is responsive to previous reviewer critiques. Nevertheless, I think there are still some points to be improved.

First, the text in Figure 1 is still blurred. It is suggested to generate a larger image size and do not enlarge the image after the figure was inserted.

Second, both references no. 1 and 22 come from the same site, they should address the same website name. Besides, the website name should not be abbreviated.

Author Response

This revised manuscript is responsive to previous reviewer critiques. Nevertheless, I think there are still some points to be improved.

First, the text in Figure 1 is still blurred. It is suggested to generate a larger image size and do not enlarge the image after the figure was inserted.

Response: Figure 1 has now been recreated using Word. It does not comprise a picture and therefore the text should now be easily legible. Please refer to p. 6 of the document.

Second, both references no. 1 and 22 come from the same site, they should address the same website name. Besides, the website name should not be abbreviated.

Response: References 1 and 22 have now been fixed. The author name has now been corrected to the Australian Commission on Safety and Quality in Health Care. These are located on p. 18, line 707 for reference 1 and p. 19, line 756 for reference 22.
